# The Union Is Strength: The Synergic Action of Long Fatty Acids and a Bacteriophage against *Xanthomonas campestris* Biofilm

**DOI:** 10.3390/microorganisms9010060

**Published:** 2020-12-28

**Authors:** Marina Papaianni, Annarita Ricciardelli, Angela Casillo, Maria M. Corsaro, Fabio Borbone, Bartolomeo Della Ventura, Raffaele Velotta, Andrea Fulgione, Sheridan L. Woo, Maria L. Tutino, Ermenegilda Parrilli, Rosanna Capparelli

**Affiliations:** 1Department of Agricultural Sciences, University of Naples Federico II, Portici, 80055 Naples, Italy; marina.papaianni@unina.it (M.P.); capparel@unina.it (R.C.); 2Department of Chemical Sciences, University of Naples Federico II, Portici, 80125 Naples, Italy; annarita.ricciardelli@unina.it (A.R.); angela.casillo@unina.it (A.C.); corsaro@unina.it (M.M.C.); fabio.borbone@unina.it (F.B.); tutino@unina.it (M.L.T.); 3Department of Physical Sciences, University of Naples Federico II, Portici, 80125 Naples, Italy; bartolomeo.dellaventura@unina.it (B.D.V.); raffaele.velotta@unina.it (R.V.); 4Istituto Zooprofilattico Sperimentale del Mezzogiorno (IZSM), Portici, 80055 Naples, Italy; andrea.fulgione@unina.it; 5Department of Pharmacy, University of Naples Federico II, Portici, 80131 Naples, Italy; sheridanlois.woo@unina.it

**Keywords:** fatty acids, *Xanthomonas*, anti-biofilm, bacteriophage, multi-target therapy

## Abstract

*Xanthomonas campestris* pv. *campestris* is known as the causative agent of black rot disease, which attacks mainly crucifers, severely lowering their global productivity. One of the main virulence factors of this pathogen is its capability to penetrate and form biofilm structures in the xylem vessels. The discovery of novel approaches to crop disease management is urgent and a possible treatment could be aimed at the eradication of biofilm, although anti-biofilm approaches in agricultural microbiology are still rare. Considering the multifactorial nature of biofilm, an effective approach against *Xanthomonas campestris* implies the use of a multi-targeted or combinatorial strategy. In this paper, an anti-biofilm strategy based on the use of fatty acids and the bacteriophage (Xccφ1)-hydroxyapatite complex was optimized against *Xanthomonas campestris* mature biofilm. The synergic action of these elements was demonstrated and the efficient removal of *Xanthomonas campestris* mature biofilm was also proven in a flow cell system, making the proposed approach an effective solution to enhance plant survival in *Xanthomonas campestris* infections. Moreover, the molecular mechanisms responsible for the efficacy of the proposed treatment were explored.

## 1. Introduction

The ability to form biofilm is a key virulence factor for a wide range of microorganisms that cause recalcitrant infections, since bacteria in biofilm show an increased tolerance towards antibiotics that leads to difficulty in eradication with conventional treatment strategies [1,2,3]. The multifactorial nature of biofilm development and drug tolerance imposes great challenges on the use of conventional antimicrobials and indicates the need for a multi-targeted or combinatorial strategy. Moreover, given the problematic spread of drug-resistant bacteria partially caused by an immoderate and negligent use of drugs, there has recently been a renewed interest in the antimicrobial effects of natural compounds, commonly used as health remedies in the past.

It is now ascertained that fatty acids possess anti-infective activities, exerted through various physical, chemical, or biological mechanisms. However, the importance of the protective roles they play in nature was recently reviewed, not only as antibacterial agents at higher concentrations but also as inhibitors of bacterial colonization and virulence at low concentrations [4].

The first evidence of lipids with antimicrobial activity was reported in 1881, when Robert Koch, testing different soap combinations, observed that FAs (Fatty Acids) were able to inhibit the growth of *Bacillus anthracis* [5]. This discovery laid the foundation for the scientific understanding of the historical role of soaps as disinfectants and detergents. Recently, many researchers have directed their interests towards the study of antimicrobial lipids and numerous studies demonstrated that FAs inhibit or kill a wide spectrum of pathogens, suppress the expression of Quorum Sensing-regulated genes, reduce swarming motility, adhesion, and virulence, and directly induce biofilm dispersion [6]. Among the pathogens sensitive to molecules that contain long fatty acid moieties, there is *Xanthomonas campestris pv. campestris* (Xcc) [6]. *Xcc* is an important bacterial pathogen responsible for black rot, one of the most problematic diseases for cauliflower and other crucifers [7]. Black rot disease causes serious damage to plants, resulting in a considerable reduction of production yield and quality of the cruciferous crop worldwide [8]. For the preventive control of this disease, some approaches have been developed, based on the exclusion of whole or parts of infected plants, or cleaning and disinfestation procedures for crop manipulation, but there are no commercial products registered to date [9]. The only available chemical strategies regard the treatment of black rot by pesticides and antibiotics. However, concerns regarding the use of antibiotics in plant disease control and its potential impacts on human health have been raised, leading to the strict limitation of antimicrobial use [10].

One of the most successful virulence factors of *Xcc* is its capability to form biofilm structures in the xylem vessels, ensuring its multiplication and dissemination throughout the plant [11]. Therefore, an efficient strategy to face an *Xcc* infection should reduce the biofilm biomass and weaken the structure at the same time as killing the bacterial cells embedded in it, so a multi-agent strategy for black rot treatment can be proposed.

Previous papers indicate that long fatty acids could interfere with the biofilm development of *Xcc* [11,12], therefore, in this paper, we tested the anti-biofilm effects of several long fatty acids in combination with a bacteriophage as an antimicrobial agent. Compared to antibiotics, bacteriophages have the advantage that their concentration increases after reaching the site of infection due to a self-replication mechanism [13]. A further advantage of using phages is their ability to grow rapidly and exponentially and therefore a single dose is enough to control the infection. Since the replication of phages occurs through bacteria, they will be eliminated once the infection has been eradicated [14]. Some limitations explain the reduced use of bacteriophages in empirical therapy [15]. One drawback in using phages could be their lack of stability over time and their low activity against some intracellular infections, associated with low pH, although recently it was demonstrated that hydroxyapatite (HA) was able to chemically interact with bacteriophages, increasing and stabilizing the activity of bacteriophages at different pH values [16].

Very recently, a lytic bacteriophage (Xccφ1) able to reduce the *Xcc* proliferation was isolated and characterized [17]; the phage Xccφ1 displayed the ability, in vivo, to reduce the symptoms of black rot disease [17]. However, the treatment of *Xcc* with phage Xccφ1 allowed only an alteration of the biofilm structure of *Xcc* [17], therefore, an effective phage therapy requires combined treatment with an anti-biofilm agent, able to completely destabilize the mature structure. In a very recent paper, a preliminary evaluation of a very promising anti-biofilm strategy that combines the use of bacteriophage Xccφ1 stabilized in a complex with hydroxyapatite (HA) and a long fatty acid was reported [12].

In this paper, the combined use of different long fatty acids and the bacteriophage Xccφ1, stabilized in a complex with hydroxyapatite (HA) [16], was optimized to eradicate *Xcc* mature biofilm. Moreover, the combined treatment with fatty acids and the bacteriophage was evaluated in flow cell system and the molecular mechanism responsible for the synergic action was explored. In detail, the structural characterization of the HA-Xccφ1 complex in the presence of fatty acid was performed and the effect of fatty acid treatment on key genes involved in Xcc quorum sensing was assessed.

## 2. Materials and Methods

### 2.1. Bacterial Strains and Culture Conditions

The bacterial strain used in this work was isolated from leaves of *Brassica oleracea* var. *gongylodes* plants with symptoms of black rot. Briefly, the leaves were washed with distilled water and disrupted mechanically, and tissue fragments were ground and streaked on mCS20ABN agar medium [18]. Yellow mucoid colonies were purified on nutrient agar (Sigma Aldrich, Milan, Italy) supplemented with glucose 0.5% (NAG). Single colonies were then suspended in sterile distilled water (SDW) and stored at 4 °C. The isolates were identified by the Biolog^TM^ System (Hayward, CA, USA) as *Xcc*. Bacteria selected after Biolog^TM^ identification were grown on nutrient agar (Merck KGaA, Darmstadt, Germany) and nutrient broth (Merck KGaA) at 24 °C. The PCR and sequencing of the 16 S were performed according to Fargier et al. [19]. In particular, the *rpoD* gene (forward primer: 5′-ATGGCCAACGAACGTCCTGC-3′, reverse: 5′-AACTTGTAACCGCGACGGTATTCG-3′) and *gyrD* gene (forward primer: 5′-TGCGCGGCAAGATCCTCAAC-3′, reverse: 5′-GCGTTGTCCTCGATGAAGTC-3′) were analyzed using PCR with the following thermal conditions: 35 cycles, each consisting of 50 s at 94 °C, 50 s at 60 °C (*rpdD*) and 62 °C (gyrD), and 1 min at 72 °C, with initial denaturation of 3 min at 94 °C and final extension of 7 min at 72 °C. The purification of PCR products was carried out using a QIAquick PCR Purification Kit (Qiagen, Hilden, Germany). The sequencing was performed using a NextSeq 500 sequencing platform by Illumina (San Diego, CA, USA).

### 2.2. Isolation and Growth of Xcc Phages

Xccφ1 was isolated from rhizospheric soil of different *Brassica oleracea* var. *gongylodes* plants with black rot symptoms [17]. In brief, the soil sediments were removed by centrifugation, and the supernatants transferred to sterile flasks. *Xcc* were incubated overnight at 24 °C in shaking conditions. Cultures were filtered through a 0.22 µm pore membrane filter (MF-Millipore, Darmstadt, Germany) and the filtrates were assayed for the presence of *Xcc*-infecting phages on a soft agar overlay. The clear plaque on soft agar containing phages was picked and incubated for 4 h at 37 °C, centrifuged for 30 min at 5000 rpm and filtered through 0.22 µm pore membrane filters [20].

The latent period and burst size for of the isolated Xccφ1 were 30 min and 42 ± 4 viral particles per infected cell, respectively [17] The phage growth curve displayed the canonical phases of latency, replication, and host lysis and the lytic activity was phage concentration independent [17].

The phages were stored at −20 °C and available in our laboratory.

### 2.3. Biomimetic HA Nanocrystal Synthesis and Characterization

Biomimetic hydroxyapatite (HA) nanocrystals were produced as described by Nocerino et al. [21]. Specifically, HA was precipitated from a solution of (CH_3_COO)_2_Ca (75 mM) by the slow addition of an aqueous solution of H_3_PO_4_ (50 mM), maintaining pH at 10 (addition of NH_4_OH). The synthesis was performed at Room Temperature (RT). Finally, the suspension of HA was washed with distilled water to remove ammonium ions and favor the interaction between nanocrystals.

### 2.4. Synthesis of Long Fatty Acids

Dodecanoic acid (C12:0), pentadecanoic acid (C15:0), and hexadecanoic acid (C16:0) were purchased from Sigma-Aldrich. The octadecanoic and eicosanoic acid (C18 and C20:0, respectively) were synthesized starting from the corresponding alcohols purchased from Sigma-Aldrich. The oxidation of the alcohols was obtained as reported by Huang et al. [22]. A solution of 1-octadecanol (30 mg, 0.15 mmol) in CH_2_Cl_2_ (3 mL) and H_2_O (650 µL) was added, in an ice-water bath, to aqueous solutions of NaBr (1 M, 95 µL), tetrabutylammonium bromide (1 M, 190 µL), (2,2,6,6-Tetramethylpiperidin-1-yl)oxyl (TEMPO, 7.03 mg, 0.45 mmol), and NaHCO_3_ (475 µL). The obtained mixture was treated with an aqueous solution of NaOCl (570 µL), kept under magnetic stirring until room temperature, and then neutralized with HCl. After neutralization, 2.66 mL of t-BuOH, (2-methyl-2-propanol, 5.32 mL of 2-methylbut-2-ene in tetrahydrofuran (THF), and a solution of NaClO_2_ (150 mg) and NaH_2_PO_4_ (120 mg) in 700 µL of water were added. After 2 h at room temperature, the mixture was diluted with a saturated aqueous solution of NaH_2_PO_4_ (15 mL) and extracted with ethyl acetate. The organic layer was then dried over Na_2_SO_4_, filtrated, and concentrated in vacuum. Then, the solution was completely dried under a stream of argon. The same procedure was utilized for the 1-eicosanol for the obtainment of the corresponding acid.

To verify the purity of the derivatives, the compounds were analyzed on an Agilent 7820 A GC System-5977B MSD spectrometer equipped with a 7693A automatic injector and an HP-5 capillary column (Agilent, flow rate 1 mL/min; He as carrier gas), using the following temperature program: 150 °C for 3 min, 150 °C to 300 °C at 15 °C/min, 300 °C for 5 min.

### 2.5. Minimum Inhibitory (MIC) and Minimum Bactericidal Concentration (MBC)Determination of Fatty Acids on Xcc

MIC was determined by the broth macro-dilution method to determine the power of anti-biofilm activity of the long fatty acids (FAs) mentioned above [23]. Serial two-fold dilutions of the working concentrations were made in a tube using Nutrient Broth (NB) (in triplicate). All controls including the sterility control (NB and deionized water), positive control (containing bacterial suspension and NB, without FAs), and the negative control (NB and DMSO) were also distributed in 2 mL sterile tubes. Each test and positive control tube was inoculated with 100 μL bacterial suspension with final concentrations of 1 × 10^7^. All experiments were performed in triplicate. In the first step, the tubes as prepared were incubated at 24 °C for 24 h under constant agitation (EW-51900-19 Stuart Equipment, Stone, UK), then 10 μL of each tube were diffused on the nutrient agar plate. The optical density of the tube was measured at 620 nm (Eppendorf BioSpectrometer^®^ basic, Milan, Italy) and the bacterial growth on the nutrient agar plate after 24 h and 48 h at 24 °C. For determining the minimum bactericidal concentration (MBC), the samples taken from each tube were spread on nutrient agar plates and incubated overnight at 24 °C. MBC was considered the concentration which corresponded to no bacterial growth. Each of these tests was performed twice (n = 2).

### 2.6. Long Fatty Acid Anti-Biofilm Activity

The biofilm formation with or without the different compounds was evaluated using crystal violet staining. All the molecules were tested at 60 µg/mL. In the experiment, 200 µL of cells were incubated in each well of sterile a 96-well flat-bottomed polystyrene plate (Falcon) for 72 h at 24 °C without shaking, to allow bacterial biofilm formation. Next, the FAs were added for 8 h. As a negative control, the untreated biofilm was used. After treatment, planktonic cells were gently removed; each well was washed with distilled water three times. To quantify the biofilm formation, each well was marked with 0.1% crystal violet and incubated for 10 min at RT, and washed with distilled water. The colorant bound to adherent cells was solubilized with 20% (*v*/*v*) acetone and 80% (*v*/*v*) ethanol. After 10 min of incubation at RT, (OD) was measured at 600 nm to quantify the total biomass of biofilm formed in each well. Each data point is composed of three independent experiments, each performed in six replicates. The different acids were screened to choose the best candidate using the same experimental procedure as above.

### 2.7. Complex HA-Xccφ1+C20:0

The HA-Xccφ1+C20:0 complex was prepared by mixing 1 mL of HA (100 mg/mL) with 1 mL of Xccφ1 (10^8^ PFU/mL) and C20:0 (60 µg/mL) and incubated—under shaking conditions—at room temperature for different times (0, 30′, 90′, 180′, 300′, and 24 h). After the proper incubation time, the sample was centrifuged, the pellet was suspended in H_2_O buffer, and the supernatant tested for phages. The concentration of the active phage particles in the pellet was evaluated by the double-layer assay (DLA) method [24]. Specifically, after overnight incubation, the phage particles were counted in order to assess the highest activity of Xccφ1. The samples that showed the best activity with the lowest concentration were selected as the optimal incubation time. At the same time, the supernatant was tested to detect the amount of phage tied to the HA. Briefly, the supernatant was spotted (3 spots of 10 µL) on the soft agar overlay. After overnight incubation, the titer was determined as reported by Papaianni et al. [24].

After that, the evaluation of the complex was carried out using all the molecules (60 µg/mL) and the activity of the phage alone (10^9^ PFU/mL), the phage and the molecules (10^9^ PFU/mL and 60 µg/mL, respectively), and the complex were compared using crystal violet staining. Two hundred microliters of *Xcc* were incubated in a sterile 96-well flat-bottomed polystyrene plate (Falcon) for 72 h at 24 °C without shaking; afterwards, the different treatments were added for 3 h and 6 h. After treatment, planktonic cells were gently removed and washed with H_2_O. Each well was marked with 0.1% crystal violet and incubated for 10 min at RT, and washed with double-distilled water. The colorant was solubilized with 20% (*v*/*v*) acetone and 80% (*v*/*v*) ethanol. After 10 min of incubation at RT, OD was measured at 600 nm to quantify the total biomass of biofilm. Each data point is composed of three independent experiments, each performed in six replicates.

### 2.8. Statistical Analysis

The statistical analysis was performed using the measured absorbance of the biofilm (triplicate) when the phages, HA, and the acids were combined among them. In particular, for this type of analysis, it was convenient to introduce the parameters *ρ* (normalization as a function of control absorbance) and *C_syn_* (synergistic coefficient).

Once defined, the *ρ* parameter was:$$\rho_{a}=1-\frac{abs_{ctrl}\;-abs_{a}}{abs_{ctrl}}$$
it was possible to estimate the synergistic coefficient *C_syn_* as:$$C_{syn}=\frac{\rho_{a}*\rho_{b}}{\rho_{ab}}$$

Moreover, for the calculation of the error analysis, we used:$$R=R\left(X,Y,\ldots \right)\;\partial R=\sqrt{{\left(\frac{dR}{dX}\cdot \partial X\right)}^{2}+{\left(\frac{dR}{dY}\cdot \partial Y\right)}^{2}+\ldots }$$

If *R* is a function of *X* and *Y*, written as *R(X,Y)*, then the uncertainty in *R* is obtained by taking the partial derivatives of *R* with respect to each variable, multiplying with the uncertainty in that variable, and adding these individual terms in quadrature.

### 2.9. Optimization of HA-Xccφ1+C20:0 Complex

The optimization of the complex was performed using crystal violet staining with different concentrations of the compounds and different times of incubation, in particular, HA (10 and 5 mg/mL), Xccφ1 (10^9^ and 10^8^ PFU/mL), and C20:0 (30 µg/mL) were tested for three and six hours. To quantify the biofilm formation, each well was stained with 0.1% crystal violet and incubated for 10 min at RT, and washed with double-distilled water. The dye bound to adherent cells was solubilized with 20% (*v*/*v*) acetone and 80% (*v*/*v*) ethanol. After 10 min of incubation at RT, OD was measured at 600 nm to quantify the amount of biofilm present in each well. Each data point is composed of three independent experiments, each performed in six replicates.

### 2.10. Confocal Laser Scanning Microscopy Analysis for Static Biofilm Evaluation

The anti-biofilm activity of the selected samples was also evaluated by confocal laser scanning microscopy (CLSM). *Xcc* biofilms were formed on Nunc^TM^ Lab-Tek^®^ 8-well chamber slides (n◦ 177445; Thermo Scientific, Ottawa, ON, Canada). Briefly, the wells of the chamber slide were filled with overnight growth of *Xcc* diluted at 0.001 (OD) 600 nm. The bacterial culture was incubated at 24 °C for 96 h to allow the *Xcc* biofilm formation. Then, the mature biofilms were incubated for 4 h in the absence and in the presence of only C20:0, and phage (10^8^ CFU/mL) plus HA (10 mg/mL) and C20:0 (30 µg/mL), in order to assess their anti-biofilm activity and their influence on cell viability. The biofilm cell viability was determined by the FilmTracer^TM^ LIVE/DEAD^®^ Biofilm Viability Kit (Molecular Probes, Invitrogen, Carlsbad, CA, USA) following the manufacturer’s instructions.

After rinsing with filter-sterilized PBS, each well of the chamber slide was filled with 300 μL of working solution of fluorescent stains, containing the SYTO^®^ 9 green fluorescent nucleic acid stain (10 μM) and propidium iodide and the red fluorescent nucleic acid stain (60 μM), and incubated for 20–30 min at room temperature, protected from light. Excess stain was removed by gently rinsing with filter-sterilized PBS.

All microscopic observations and image acquisitions were performed with a confocal laser scanning microscope (CLSM; LSM700-Zeiss, Jena, Germany) equipped with an Ar laser (488 nm), and a He-Ne laser (555 nm). Images were obtained using a 20×/0.8 objective. The excitation/emission maxima for these dyes are 480/500 nm for SYTO^®^ 9 stain and 490/635 nm for propidium iodide. Z-stacks were obtained by moving the microscope to a point just out of focus on both the top and bottom of the biofilms. Images were recorded as a series of tif files with a file size of 16 bits.

### 2.11. Confocal Laser Scanning Microscopy for Dynamic Biofilm Evaluation

The effect of the selected samples on *Xcc* biofilm was evaluated using flow cell methods, that allow non-invasive and non-destructive examination of biofilms. A parallel analysis of *Xcc* biofilms was performed using a 3-channel flow cell chamber (IBI Scientific. Peosta, IA, USA). For this study, a solution of phosphate-buffered saline (PBS, pH 7) was flowed into each channel of the cell at a controlled flow rate of 160 µL/min using an Ismatec™ IPC 4 Peristaltic Pump (Cole-Parmer GmbH, Wertheim, Germany). The flow system was kept free of air bubbles using a bubble trap, which created a low positive pressure with medium solution flow, thus mitigating undesirable peristaltic pulsation in liquid delivery to the flow cell. Then, a bacterial suspension of *Xcc* at 0.5 OD was circulated through the system for 2 h and the non-adhering cells were washed away with sterile PBS for 15 min. Finally, fresh medium (nutrient broth 50% *v*/*v* in PBS) was circulated for 48 h through the system to let the biofilm form. After incubation, three different solutions were circulated for 3 h into each channel of the cell: only fresh medium into the first channel (NT), fresh medium containing only phage (10^8^ CFU/mL) into the second channel, and fresh medium containing the HA-Xccφ1+C20:0 complex into the third channel. Biofilms formed in the absence and presence of the tested samples were evaluated by confocal laser scanning microscopy (CLSM). Polydimethylsiloxane(PDMS) surfaces were evaluated by confocal laser scanning microscopy (CLSM). The biofilm cell viability was determined by the FilmTracer™ LIVE/DEAD^®^ Biofilm Viability Kit (Molecular Probes, Invitrogen, Carlsbad, CA, USA) following the manufacturer’s instructions. In detail, a volume of 500 µL of a working solution of fluorescent stains, containing the SYTO^®^ 9 green fluorescent nucleic acid stain (10 µM) and propidium iodide and the red fluorescent nucleic acid stain (60 µM) was injected with a syringe into each channel, without removing the flow cell from the flow system, and incubated for 20–30 min at room temperature, protected from light. Then, fresh PBS was used to remove the excess stain. All microscopic observations and image acquisitions were performed as previously described.

### 2.12. SEM Image

Water suspensions of the samples HA-Xccφ1+C20:0 complex, HA (10 mg/mL), Xccφ1 (10^8^ PFU/mL), and C20:0 (30 µg/mL)—previously centrifuged at 13,000 rpm for 15 min—were deposited on 5 × 5 mm silicon chips and the solvent was evaporated under vacuum at 30 °C. The silicon supports were mounted on 13 mm SEM aluminium stubs and sputtered with a nanometric conductive layer of Au/Pd alloy using a Desk V TSC coating system (Denton Vacuum, Moorestown, NJ, USA). SEM micrographs were recorded with a Nova NanoSem 450 field emission gun scanning electron microscope (FEGSEM) (FEI/Thermofisher, Hillsboro, OR, USA), under high-vacuum conditions.

### 2.13. Z-Potential

Water suspensions of the samples HA-Xccφ1+C20:0 complex, HA (10 mg/mL), Xccφ1 (10^8^ PFU/mL), and C20:0 (30 µg/mL) were analyzed for the measurement of zeta potential in disposable folded capillary cells (Malvern Instruments, DTS1070, Malvern, UK) using a Zetasizer Nano ZS (Malvern Instruments). Each analysis was carried out in triplicate for three independent experiments. The analysis temperature was 25 °C and about 1 mL of sample was used for the test. The results were analyzed, and for each sample, the zeta potential average value was determined.

### 2.14. RNA Extraction and Expression Profiling by qPCR

Fifty milliliters of bacterial cells were incubated in a narrow mouth Erlenmeyer flask for for 72 h at 24 °C in a static manner for biofilm formation and then phage (10^8^ PFU/mL), C20:0 (30 µg/mL), and HA-Xccφ1+C20:0 were added. Every 30 min, 10 mL of biofilm were taken for 2 h. Total RNA was extracted using a TRIzol protocol [25]. A NanoDrop^®^ ND-1000 (Thermo Fisher Scientific Inc., Waltham, MA, USA) was used to assess total RNA quantity. One microgram of purified total RNA was used as a template for first-strand cDNA synthesis using SuperScript^®^ III Reverse Transcriptase (Invitrogen), where the first step of the protocol is DNase I treatment (37 °C, 20 min) followed by a DNase I inactivation/+EDTA (65 °C, 10 min). The primers were designed using https://www.eurofinsgenomics.eu/en/ecom/tools/qpcr-assay-design/ for all genes: *clp* (Fw5′-GACGGGAAAGGGGGCAATTA-3′; Rw5′-CACAACCGTCGTGTTCCCTA-3′), *manA* (Fw5′-CACCTTCCGCAGCAACAATC-3′; Rw5′-AGCACGATATCCAGCGACTG-3′), *rpfF* (Fw5′-CGACGCTTTCCAATAGCACG-3′; Rw5′-AGCGTCGATACCTGCTGATG-3′), *gumB* (Fw5′-GGTTCGACCTGACCGAGATC-3′; Rw 5′-CGCCGCCATAAATCTCAGGA-3′). All controls are reported in Appendix A.

Gene transcript levels were measured using Power SYBR^®^ Green PCR Master Mix (Applied Biosystems^®^) on a QuantStudio™ 3 Real-Time PCR System (Applied Biosystems^®^, Thermo Fisher Scientific, Waltham, MA, USA) with the following conditions: an initial step at 95 °C for 10 min, followed by 40 cycles of 95 °C for 15 s, *clp* at 57.1 °C; *manA* at 55 °C; *rpfF* at 59.9 °C; *gumB* at 63.7 °C for 40 s, and 72 °C for 1 min. QuantStudio Design and Analysis Software v1.1 (Applied Biosystems) was used for the analysis of gene expression. All samples were normalized to *rpoB* as a housekeeping gene. The relative quantitative expression was determined using the 2^−ΔΔCT^ method [26].

Phage Xccφ1 titration

Briefly, 200 µL of *Xcc* were incubated in a sterile 96-well flat-bottomed polystyrene plate (Falcon) for 72 h at 24 °C without shaking; afterwards, phage (10^8^ PFU/mL), phage with HA (10^8^ PFU/mL and 100 mg/mL, respectively) and the HA-Xccφ1+C20:0 complex were added to the biofilm for 6 h. Biofilm was removed, vortexed, and treated with chloroform for 10 min to eradicate all the bacteria. The estimation of the active phage particles in the biofilm was evaluated by the double layer assay (DLA) method [24]. The titration of the phage after the treatment was performed with a double layer assay using the following equation:PFU/mL = n° of lysis plates/n° of spots for dilution * dilution

## 3. Results

### 3.1. The Effect of Saturated Long Fatty Acids on Xcc Biofilm Eradication

The MIC and the MBC of FAs on *Xcc* were investigated, and purified FAs did not possess antibacterial activity on Xcc cells at the tested concentrations (from 40 µg/mL to 300 µg/mL). The anti-biofilm activity of different saturated long-chain fatty acids (pentadecanoic (C15:0), dodecanoic (C12:0), hexadecanoic (C16:0), octadecanoic (C18:0), and eicosanoic (C20:0) acid) were tested on *Xcc* mature biofilm. To assess if the selected molecules were able to eradicate a preformed biofilm, a 72 h mature *Xcc* biofilm was treated with the different acids for 8 h at 25 °C. The results indicated a clear reduction of *Xcc* mature biofilm after the treatment with all the tested molecules, with the highest eradication percentages in the case of the acids C16:0, C18:0, and C20:0 (Figure 1A).

### 3.2. Synergic Treatment of Mature Xcc Biofilm with HA–Xccφ1 and Long-Chain Fatty Acids

The anti-biofilm activity of the different long-chain fatty acids (C12:0, C15:0, C16:0, C18:0, and C20:0) was evaluated in the presence of the HA-Xccφ1 complex on *Xcc* mature biofilm. In particular, the anti-biofilm efficacy of long-chain fatty acids was evaluated in combination with Xccφ1 alone or the HA-Xccφ1 complex. The results demonstrated that the addition of the HA-Xccφ1 complex resulted in a more effective anti-biofilm treatment, regardless of the acid used (Figure 1B).

Moreover, to assess if the effect induced by the simultaneous treatment with the HA– Xccφ1 complex and fatty acids was additive or synergistic, the results reported in Figure 1B were analyzed using a statistical approach. Two parameters (ρ and C_syn_), described in Section 2.8, were used to perform this evaluation. Taking into account that high C_syn_ values correspond to a clear synergistic effect, as reported in Table 1, the best synergistic effect was obtained when the anti-biofilm treatment was performed using C20:0 and the HA-Xccφ1 complex, even if a synergistic behavior was also recorded in the presence of C12:0 and C18:0 (Table 1).

The combination of C20:0 and HA-Xccφ1 (HA-Xccφ1+C20:0) was selected for subsequent experiments. In particular, the optimization of the treatment with C20:0 and the HA-Xccφ1 complex was performed by modifying the concentrations of C20:0, Xccφ1, and HA (Figure 2), and exploring different incubation times (data not shown). The optimal condition to eradicate *Xcc* mature biofilm was obtained using Xccφ1 (10^8^ CFU/mL), HA (5 mg/mL), and C20:0 (30 µg/mL) in a 3 h treatment.

### 3.3. Characterization of the Anti-Biofilm Activity of Combined HA–Xccφ1 and C20:0 Treatment on Xcc Mature Biofilm

The anti-biofilm activity of C20:0 alone and in combination with HA-Xccφ1 on *Xcc* mature biofilm was evaluated by confocal laser scanning microscopy (CLSM). In detail, 72 h mature biofilm of *Xcc* was incubated in the presence of HA-Xccφ1+C20:0. As shown in Figure 3A, although C20:0 led to a significant reduction in the biofilm biomass, a stronger biofilm-inhibiting effect was obtained when it is used in combination with HA-Xccφ1.

The anti-biofilm effect of HA-Xccφ1+C20:0 treatment on *Xcc* mature biofilm was also explored in dynamic conditions, using a flow cell system. Three-channel flow cells were used for the analysis of *Xcc* biofilm formation, in the absence and presence of C20:0 or HA-Xccφ1+C20:0. Mature biofilms were treated for 3 h, under a flow rate of 160 µL/min, with only fresh medium, or with C20:0 or HA-Xccφ1+C20:0. After treatment, the biofilms were observed by CLSM (Figure 3B). Although a substantial reduction in biofilm biomass was obtained in both cases, the multi-agent treatment resulted in a more evident reduction of *Xcc* mature biofilm. To obtain more detailed information on the biofilm structure, the collected three-dimensional images were analyzed using the COMSTAT image analysis software package. An increased roughness coefficient is observed for treated samples (Appendix A); this dimensionless factor provides a measure of how much the thickness of a biofilm varies, and it is thus used as a direct indicator of biofilm heterogeneity. The analysis revealed that the treatment with HA-Xccφ1+C20:0 led to the formation of inhomogeneous and unstructured biofilms (Appendix A).

### 3.4. Characterization of the HA–Xccφ1 Complex in the Presence of C20:0

In order to investigate how and if Xccφ1, HA, and C20:0 interact with each other, scanning electron microscopy (SEM) analysis was carried out. In particular, SEM images of pristine HA show porous spherical aggregates of elongated crystallites, with variable dimensions from a few hundred nanometers to a few microns (Figure 4A). When HA is functionalized with C20:0 acid, the porous structure appears to be filled with the organic acid and only the external shape of the aggregates is visible (Figure 4B). In the case of HA-Xccφ1, the porous structure of HA is still visible, although partially filled. A deeper investigation of the particles’ surface showed HA crystallites were effectively functionalized with Xccφ1, as can be appreciated by the comparison of inset A and C of Figure 4, demonstrating that HA can be complexed with Xccφ1. A similar level of functionalization was observed in the case of the HA-Xccφ1+C20:0 complex (Figure 4D). This observation was confirmed by assessing the amount of Xccφ1 adsorbed on HA in the presence and absence of C20:0, indeed, the amount of bacteriophage adsorbed on HA in the two conditions was comparable (Appendix A).

Moreover, zeta potential measurements were performed for HA, HA-Xccφ1, Xccφ1, and HA-Xccφ1+C20:0 to investigate the change of the surface charge of the different samples. The results confirmed the interaction of the phage Xccφ1 with hydroxyapatite, obtaining a zeta potential value of the complex of about −3 mV compared to only HA and only phage, with values of 0 mV and −15 mV, respectively (Figure 5). Regarding only HA treated with C20:0, no evidence of the modification of surface charge was observed (0 mV). Thus, the presence of C20:0 affects, even if only small changes, the surface charge of the HA. The peak in the case of the HA-Xccφ1+C20:0 complex corresponds to positive values (4 mV), demonstrating that C20:0 did not interfere with the formation of the complex between HA and Xccφ1.

### 3.5. Effect of C20:0 Treatment on Genes Involved in Xcc Quorum Sensing

In order to collect information on the molecular mechanisms involved in the anti-biofilm activity of C20:0, the expression of some key genes in quorum sensing and biofilm formation of *Xcc* was investigated. In particular, the gene expression of *rpf**F*, *gumB*, *clp*, and *manA* was evaluated using the Ct method. The *rpfF* gene encodes a enoyl-CoA hydratase that is involved in the synthesis of diffusible signal factor (DSF) [27]; *gumB* encodes an outer membrane xanthan exporter and is essential for xanthan biosynthesis [28]; the *clp* gene encodes a global regulator (cAMP receptor protein-like protein) involved in the *Xcc* QS regulatory pathway [29], and the *manA* gene encodes an endo-1,4-mannanase [30]. The absolute Ct values from the qPCR assays were used to calculate the expression ratios of the *rpf**F* [27], *gumB* [28], *clp* [29], and *manA* [30] genes in *Xcc* biofilm cells treated or not treated with C20:0. The relative gene expression was normalized to *rpoB* as a reference housekeeping gene. The expression of *gumB* is not affected by the treatment with C20:0 and *rpfF* is up-regulated, while *clp* is downregulated, in the presence of C20:0 (Figure 6). The *manA* gene expression was deeply influenced by C20:0 treatment, indeed, a clear overexpression of this gene was recorded after 60 min of treatment.

## 4. Discussion

The formation of biofilm is a crucial process for the survival and persistence of bacteria in most habitats on Earth. For plant-associated bacteria, the formation of biofilms has evolved as an adaptive strategy to successfully achieve host colonization and as a key strategy for pathogenesis [31]. Biofilm development contributes to the virulence of phytopathogenic bacteria through various mechanisms, including the blockage of xylem vessels [32].

Therefore, a possible treatment for black rot disease might be devoted to the eradication of *Xcc* biofilm and could be obtained by combining the use of antimicrobials and anti-biofilm agents. Recently, a lytic bacteriophage (Xccφ1) able to control the disease caused by *Xcc* in *Brassica oleracea var. gongylodes* was isolated and characterized [17]; the phage-based treatment was able to limit the bacterial proliferation [17]. In this paper, the phage-based treatment was improved using hydroxyapatite to increase and stabilize the activity of the bacteriophage. The complex formation of Xccφ1 with HA, demonstrated by zeta potential measurements, was effective against other pathogens [16] and, in the present work, the use of HA increased the anti-biofilm activity of Xccφ1 (Figure 1B). Following the idea of a multi-target strategy, in a very recent paper [12], treatment with the HA-Xccφ1 complex was combined with the use of a long fatty acid, eicosanoic acid. In this paper, the research was extended to different saturated long-chain fatty acids looking for the one that, in combination with HA-Xccφ1, was able to eradicate the mature biofilm of *Xcc*, and the results confirm the activity of eicosanoic acid and reveal that other long fatty acids are also endowed with anti-biofilm activity. Therefore, a statistical analysis was performed to identify the best candidate for the subsequent optimization experiments. The best synergistic effect was obtained when the treatment was performed using eicosanoic acid (C20:0) in combination with the HA-Xccφ1 complex, then the treatment was optimized by changing the incubation time and the amounts of C20:0, HA, and Xccφ1 to identify the best condition for the eradication of *Xcc* mature biofilm.

Since biofilm evolution in flow conditions is more closely related to natural biofilms, which can differ from those obtained in static biofilms formation assays, the anti-biofilm activity of HA-Xccφ1+C20:0 was also demonstrated in a flow cell system. Moreover, the CLSM analyses on treated biofilm revealed that the synergic action of HA-Xccφ1+C20:0 not only reduces the biofilm biomass but also deeply modifies the *Xcc* biofilm structure.

To collect information on the molecular mechanism responsible for the synergic anti-biofilm effect of HA-Xccφ1+C20:0 treatment, SEM images of HA samples in the presence of Xccφ1 or C20:0 or Xccφ1+C20:0 were recorded and compared. The investigation confirmed the previously reported possibility to functionalize the HA crystallites with bacteriophages [16]. The SEM analysis indicated a similar level of functionalization in the case of HA-Xccφ1 in the presence or in the absence of C20:0, suggesting that the synergic effect is not related to a different functionalization yield.

The chemical structure of C20:0 is similar to that of diffusible signal factor (DSF) molecules [33], indeed, DSF family signals share a fatty acid carbon chain with variations in chain length, double-bond configuration, and side-chains [34]. This structural similarity pointed out a possible interference of C20:0 in the *Xcc* quorum sensing system as a potential mechanism responsible for the C20:0 anti-biofilm effect and for the synergy between HA-Xccφ1 and C20:0.

The synthesis and perception of the DSF signal require products of the Rpf (regulation of pathogenicity factors) gene cluster. The *rpfF* gene encodes a key enzyme necessary for DSF biosynthesis, whereas RpfC and RpfG are a two-component system involved in signal perception and transduction [35,36]. At low bacterial cell density, RpfC binds and represses the DSF synthase of RpfF, preventing the production of DSF [37]. At high cell density, a high concentration of extracellular DSF activates the RpfC–RpfG system to degrade the bis (3′, 5′)-cyclic diguanosine monophosphate (cyclic di-GMP; c-di-GMP) [38] and, in particular, RpfG has phosphodiesterase activity able to degrade cyclic di-GMP (c-di-GMP). The c-di-GMP is an inhibitory ligand of the global transcription factor Clp [39] and, consequently, derepressed Clp drives the expression of several hundred genes [36]. The global regulator Clp, among others, positively regulates the transcription of the *manA* gene [30], encoding the extracellular enzyme beta (1,4)-mannanase implicated in biofilm dispersal [40].

In this paper, the expression in *Xcc* cells of the key genes *manA*, *rpfF*, *clp*, and *gumB* was evaluated in the presence and absence of C20:0. Results reported in the present paper demonstrated that, 60 min after the treatment of *Xcc* mature biofilm with C20:0, the transcription of the *manA* gene is induced, indicating that, in the described experimental conditions, C20:0 works as a DSF-like molecule, which by the DSF/rpf system and the regulator Clp, induces the expression of *manA* [27]. A previous paper [41] demonstrated that DSF-like molecules may influence bacterial antibiotic susceptibility in multiple ways [41], including modulation of the biofilm formation, although it was not reported if their functionality is related to their chemical properties or associated with their potential roles in the interference of bacterial signaling and regulatory networks. Instead, in this paper, we demonstrated that C20:0 acts as a DSF-like molecule interfering with the transcription of *manA*, a gene coding for an enzyme involved in the biofilm dispersion stage. Indeed, the DSF-inducible enzyme, endo-β-1,4-mannanase ManA, is known to be able to disperse the mature biofilm of *Xcc* [22]. The action of ManA could explain the anti-biofilm effect of C20:0 and the ability of this molecule to indirectly modify the *Xcc* biofilm structure (Figure 3 and Appendix A), which was more heterogeneous and unstructured when treated with C20:0.

In support of the hypothesis that C20:0 works as a DSF-like molecule, a slight effect on *rpfF* and *clp* was expected. Indeed, DSF molecules have no strong effect on *rpfF* and *clp* gene expression, since the DSF synthesis in *Xcc* is not auto-induced [42], and the influence of DSF molecules on *clp* is not transcriptional [43].

As regards the *gumB* gene, it is essential for the production of xanthan [28,44], one of the exopolysaccharides produced by *Xcc*, and it has been suggested that *clp* could control xanthan synthesis by directly binding to the *gumB* promoter [45]. On the contrary, a more recent paper demonstrated that, in *Xcc, gumB* is under the control of HpaR1, a regulator belonging to the GntR family [39] that positively regulates the production of extracellular enzymes and xanthan production. In the experimental conditions used in this paper, C20:0 seems somewhat involved in *gumB* regulation, although further experiments will be necessary to clarify this point.

In conclusion, the idea suggested to explain the reported synergy between the HA-Xccφ1 complex and C20:0 is that the C20:0 treatment induces the synthesis of *manA*, which is able to hydrolyze mannan, promoting the *Xcc* mature biofilm dispersion, and thus making the biofilm more susceptible to the antimicrobial action exerted by the HA-Xccφ1 complex on *Xcc* cells [12]. The combined use of the anti-biofilm molecule C20:0 and a bacteriophage–hydroxyapatite complex results in very effective removal of *Xcc* mature biofilm.

Although bacteriophage use in therapy is limited by the specificity of the phage–bacterium interaction and because precise identification of the pathogenic bacteria is necessary to select an appropriate bacteriophage, the proposed strategy has several advantages. Indeed, the virus replication at the site of infection allows for obtaining a high concentration of phages in the biofilm (Appendix A), increasing the antimicrobial treatment efficiency. Moreover, since viruses can infect persistent cells and remain with them until they become metabolically active [46,47], the proposed approach should reduce the occurrence of recurrent infections.

## Figures and Tables

**Figure 1 microorganisms-09-00060-f001:**
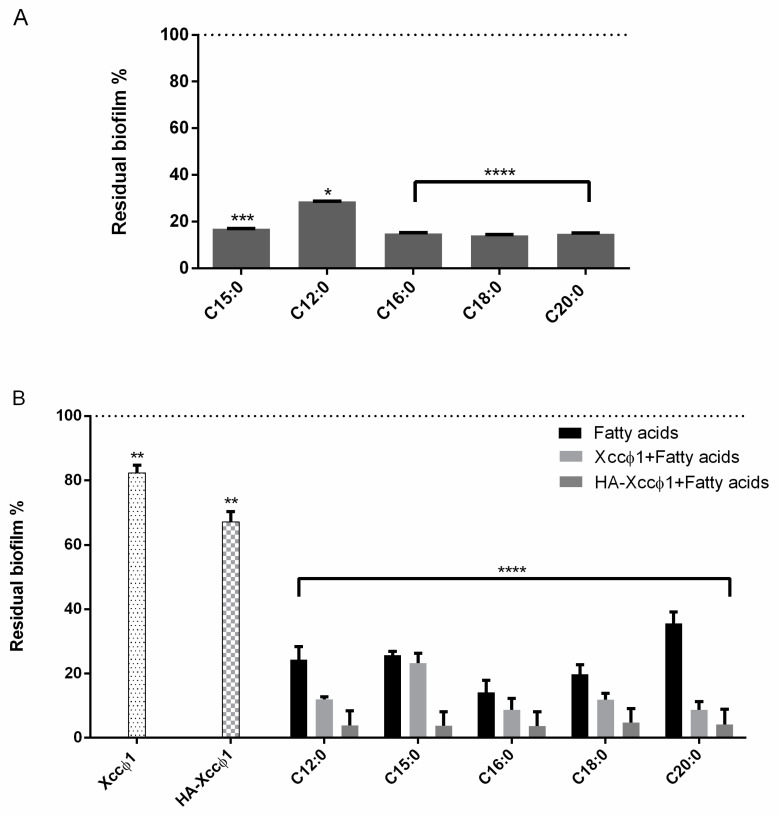
(**A**) Analysis of the effect of different fatty acids (60 µg/mL) on *Xcc* mature biofilm structure after 72 h of incubation at 25 °C and 8 h of treatment. The data are reported as percentages of residual biofilm. Each value is the mean ± SD of three independent experiments. Statistical analysis was performed with the absorbance compared to the untreated control and considered statistically significant when *p* < 0.05 (* *p* < 0.05, ** *p* < 0.01, *** *p* < 0.001, **** *p* < 0.0001) according to two-way ANOVA multiple comparisons. (**B**) Analysis of the effect of all the acids fatty acids (60 µg/mL) with Xccφ1 or Xccφ1 plus HA on *Xcc* biofilm structure using crystal violet assay after 72 h of incubation at 25 °C and 6 h of treatment. The data are reported as percentages of residual biofilm. Each value is the mean ± SD of three independent experiments. Statistical analysis was performed with the absorbance compared to the untreated control and considered statistically significant when *p* < 0.05 (* *p* < 0.05, ** *p* < 0.01, *** *p* < 0.001, **** *p* < 0.0001) according to two-way ANOVA multiple comparisons.

**Figure 2 microorganisms-09-00060-f002:**
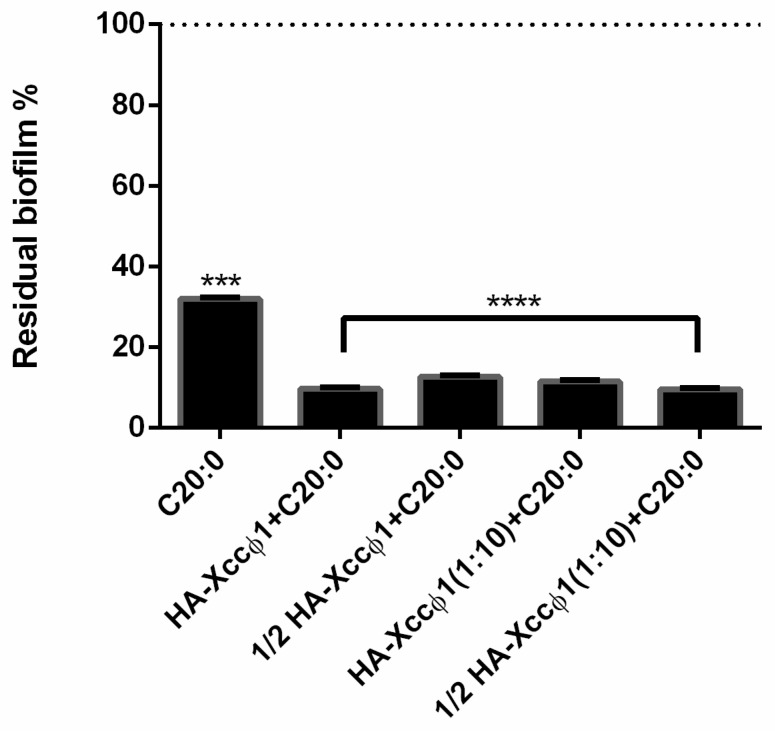
Analysis of the effect of C20:0 alone (30 µg/mL) or with Xccφ1 (φ) 10^8^/10^9^ PFU/mL or Xccφ1 plus HA on *Xcc* biofilm structure using crystal violet assay after 72 h of incubation at 25 °C and 3 h of treatment. The data are reported as percentages of residual biofilm. Each value is the mean ± SD of three independent experiments. Statistical analysis was performed with the absorbance compared to the untreated control and considered statistically significant when *p* < 0.05 *** *p* < 0.001, **** *p* < 0.0001) according to two-way ANOVA multiple comparisons.

**Figure 3 microorganisms-09-00060-f003:**
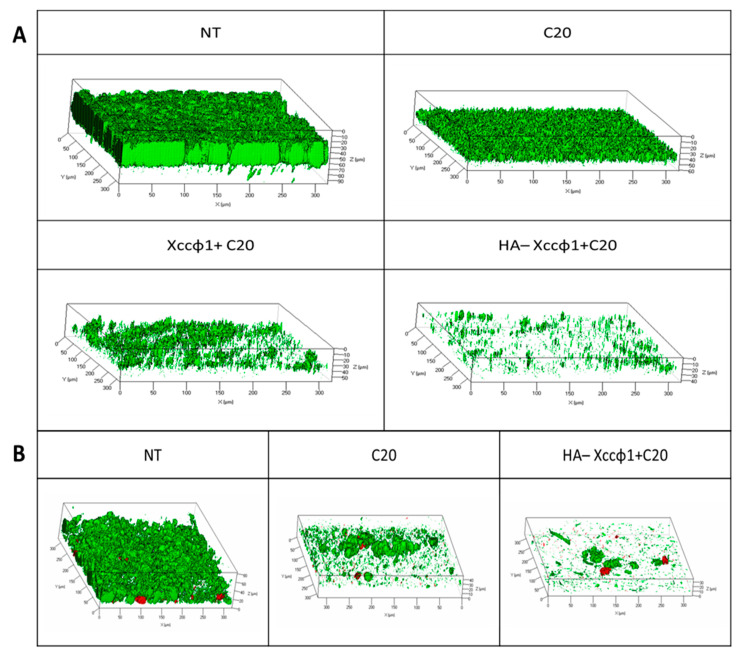
(**A**) Analysis of the anti-biofilm effect of eicosanoic acid (C20:0) alone, with Xccφ1, and in combination with Xccφ1 and HA on *Xcc* mature biofilm. CLSM analysis was performed on 72 h mature *Xcc* biofilms after 3 h of treatment at Room Temperature. The biofilm structures were stained using the LIVE/DEAD^®^ Biofilm Viability Kit. (**B**) Evaluation of biofilm formation of *Xcc* in dynamic conditions in the presence and absence of C20:0 and the complex of HA-Xccφ1+C20:0. Biofilm formation was performed in a three-channel flow cell for 48 h. CLSM analysis was performed after 3 h incubation at RT in the presence and the absence of the anti-biofilm samples.

**Figure 4 microorganisms-09-00060-f004:**
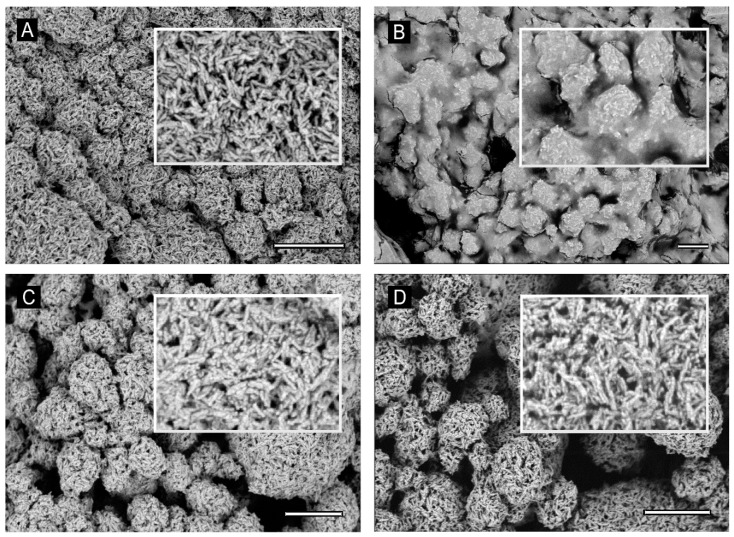
SEM image of HA 80,000x (**A**), HA + C20 35,000x (**B**), HA-Xccφ1 68,000x (**C**), and HA-Xccφ1+C20 75,000x (**D**). Scale bar is 1 µm.

**Figure 5 microorganisms-09-00060-f005:**
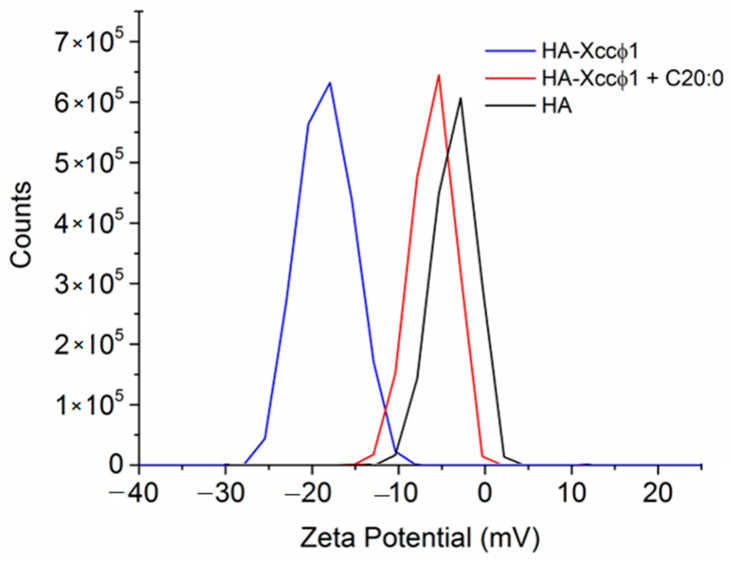
Zeta potential analysis of HA, HA-Xccφ1, and HA-Xccφ1+C20.

**Figure 6 microorganisms-09-00060-f006:**
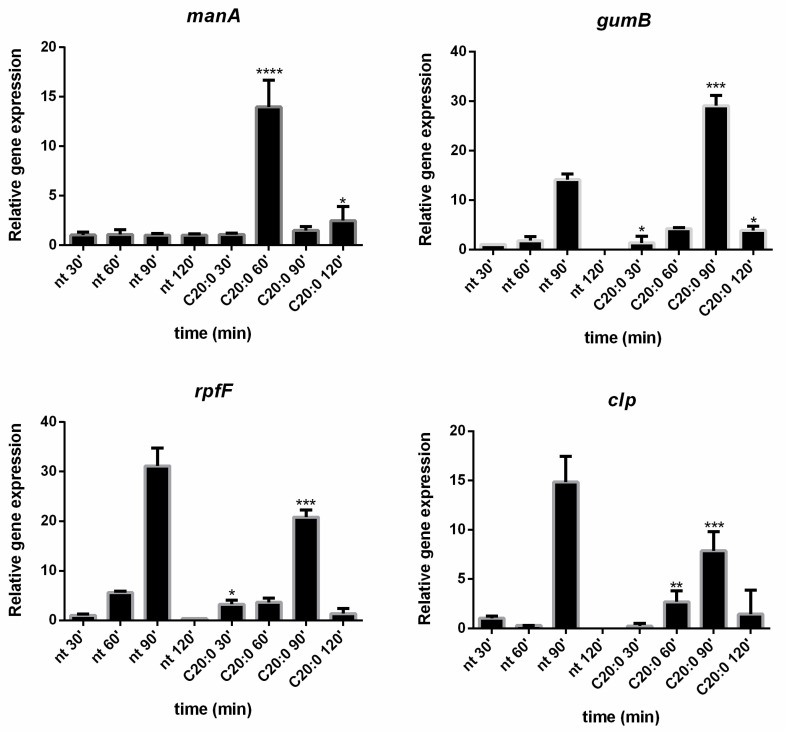
Expression profiling of *Xcc* genes by quantitative real-time PCR (qPCR) of the different genes involved in biofilm formation and quorum sensing. Biofilm samples were collected at different durations of treatment with C20:0. Statistical analysis was performed with the relative gene expression compared to the untreated control and considered statistically significant when *p* < 0.05 (* *p* < 0.05, ** *p* < 0.01, *** *p* < 0.001, **** *p* < 0.0001) according to two-way ANOVA multiple comparisons.

**Table 1 microorganisms-09-00060-t001:** The values (C_syn_) are the result of the mathematical analysis used to obtain the combination of acids, phage, and HA showing the best synergistic effect.

	C12:0	C15:0	C16:0	C18:0	C20:0
Xccφ1	2.97 ± 1.33	1.31 ± 0.61	0.84 ± 0.34	1.59 ± 0.91	1.28 ± 0.53
Xccφ1+HA	7.92 ± 3.21	1.31 ± 0.56	1.58 ± 0.69	5.0 ± 2.0	7.94 ± 3.22

## Data Availability

The data presented in this study are available within the article.

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
