# Peer review of "The Union Is Strength: The Synergic Action of Long Fatty Acids and a Bacteriophage against Xanthomonas campestris Biofilm"

_microorganisms, 2020, doi:10.3390/microorganisms9010060_

Round 1

Reviewer 1 Report

The article "the union is strength ...." proposed by Papaianni et al, deals with the effectiveness of a fatty acid/phage/hydroxyapatite complex as a new means of control of Xcc.

The authors isolated a rhizosphere soil phage grown with Brassica oleacera showing symptoms of black rot in a previous publication. In this paper, the authors produced nanocrystals of hydroxyapatite ()HA as well as 2 fatty acids, three other fatty acids were purchased. The effect of the fatty acids alone or in complex with HA and the previously isolated phage was tested on the formation of biofilm in polystyrene plate and then one fatty acid was selected to be tested in complex in CLSM (static and dynamic). Finally several Xcc genes involved in quorum sensing were followed by qPCR on a culture of Xcc in contact with the complex.

I have two majors comments:

1/ the results with the acids alone in figure 1B are very different from those in figure 1A, yet it seems that there was very little variability between trials in figure 1A, how do you explain this discrepancy?

2/ invalid qPCR :

with qPCR, it is very important to verify that there is no genomic DNA contamination. According to the materials and methods, no treatment with a DNase has been done. In addition, the effectiveness of PCR has not been verified for each of the primer pairs drawn. And finally, the reference gene selected for "hcrC" is not a gene commonly used for calibration, no justification is given for the choice of this gene.

I ask the authors to apply the MIQE guidelines: Bustin, S. A., Benes, V., Garson, J. A., Hellemans, J., Huggett, J., Kubista, M., ... & Vandesompele, J. (2009). The MIQE Guidelines: M inimum I nformation for Publication of Quantitative Real-Time PCR Experiments.

minor comments:

Part 2.2 can be reduced since it uses the method of reference 17.

L84 "prelaminar" : replace with preliminary

L108 "Illumnia" : replace with Illumina

L118 "aviable" : do you mean available?

L319 and 346 'Chrystal" : replace with crystal. In the article the words "crystal violet" are written in different ways, with or without capitalization. please homogenize them.

L408 :rpfF is underlined

Figure 1 :  replace "molecules" with fatty acids. Specify the concentration of the acids in figure 1B

Author Response

Response to Reviewer 1 Comments

Reply to Reviewer comments:

For clarity, in the response letter, each reply is reported in red, whereas in the manuscript all changes have been highlighted by using red coloured text.

Comments and Suggestions for Authors

The article "the union is strength ...." proposed by Papaianni et al, deals with the effectiveness of a fatty acid/phage/hydroxyapatite complex as a new means of control of Xcc.

The authors isolated a rhizosphere soil phage grown with Brassica oleacera showing symptoms of black rot in a previous publication. In this paper, the authors produced nanocrystals of hydroxyapatite ()HA as well as 2 fatty acids, three other fatty acids were purchased. The effect of the fatty acids alone or in complex with HA and the previously isolated phage was tested on the formation of biofilm in polystyrene plate and then one fatty acid was selected to be tested in complex in CLSM (static and dynamic). Finally several Xcc genes involved in quorum sensing were followed by qPCR on a culture of Xcc in contact with the complex.

I have two majors comments:

1/ the results with the acids alone in figure 1B are very different from those in figure 1A, yet it seems that there was very little variability between trials in figure 1A, how do you explain this discrepancy?

Re:  We thank the reviewer for this comment that allowed us to correct an error present in the legend of figure 2. Indeed as reported in material and method Paragraph (line 189 of the old version) the anti-biofilm experiment performed in presence of  Xccɸ1 and HA-Xccɸ1 complex were performed for 3 or 6 hours, in the experiment presented in figure 1 B the treatment was performed for 6 h.

2/ invalid qPCR :

with qPCR, it is very important to verify that there is no genomic DNA contamination. According to the materials and methods, no treatment with a DNase has been done. In addition, the effectiveness of PCR has not been verified for each of the primer pairs drawn. And finally, the reference gene selected for "hcrC" is not a gene commonly used for calibration, no justification is given for the choice of this gene.

Re:We thank the reviewer for these comments that allowed us to improve the quality of the material and method section.

the cDNA synthesis was performed by SuperScript® Reverse Transcriptase (Invitrogen) kit,  the first step of the protocol is DNase I treatment (37°C, 20 min) followed by a  DNase I inactivation/+EDTA (65°C, 10 min).

This information was introduced in the material and methods section.

As for  the assessment of the effectiveness of PCR with the primer pairs drawn it was verified before the qPCR experiment (if required the authors agree to the introduction of the pcr control experiments as supplementary information)  

The hcrC gene was chosen for calibration since it is previously used for the detection and identification of the crucifer pathogen, Xanthomonas campestris pv. Campestris ( Zaccardelli M., Campanile F., Spasiano A., Merighi M. (2007). Eur. J. Plant Pathol. 118 299–306. 10.1007/s10658-007-9115-y). Orthologs of HrcC are present in every functional type III secretion system, representing the basal component of the injectisomeS.  (Type III protein secretion systems in plant and animal pathogenic bacteria. Annual Review of hytopathology, 36, 363–392.).

If in the experience of the review, that likely is an expert of Xcc, the selected gene is not suitable for the qPCR calibration the authors agree to repeat the qPCR using a different calibration gene.

minor comments:

Part 2.2 can be reduced since it uses the method of reference 17.

Re: The text was modified as suggested, some information was deleted but other sentences are introduced to respond to Editor suggestions.

L84 "prelaminar": replace with preliminary

L108 "Illumnia": replace with Illumina

L118 "aviable": do you mean available?

L319 and 346 'Chrystal" : replace with crystal. In the article the words "crystal violet" are written in different ways, with or without capitalization. please homogenize them.

L408 :rpfF is underlined

 Re: The text was modified as suggested.

Figure 1 :  replace "molecules" with fatty acids. Specify the concentration of the acids in figure 1B

Re: a new version of Figure 1 was prepared

Reviewer 2 Report

The presented manuscript is a well-thought-out, properly designed and properly discussed experimental work that can be applied in the fight against X. campestris. The only reservation that would be worth considering in the discussion, or rather in the conclusion, is the issue related to the practical application of phage therapies (advantages were discussed but disadvantages were not mentioned). I suggest correcting a few typos in the text, e.g. line 443

Author Response

Comments and Suggestions for Authors

The presented manuscript is a well-thought-out, properly designed and properly discussed experimental work that can be applied in the fight against X. campestris. The only reservation that would be worth considering in the discussion, or rather in the conclusion, is the issue related to the practical application of phage therapies (advantages were discussed but disadvantages were not mentioned). I suggest correcting a few typos in the text, e.g. line 443

Re: We are deeply indebted to the Reviewer for this suggestion, a sentence underlying the disadvantages of phage therapies was introduced in the conclusion.

Reviewer 3 Report

The authors present a manuscript in which they showed the synergic action of long fatty acids and a bacteriophage against Xanthomonas  campestris biofilm. They showed the combined use of  the anti-biofilm molecule C20:0 and a bacteriophage-hydroxyapatite complex results in effective removal of Xanthomonas campestris mature biofilm.

The paper is clear and the conclusions are congruent with the aims of the work.

Minor comment:

Figure 4. 

What was magnification used?

It is not visible in the photos, please complete the legend.

2.6. Long fatty acids anti-biofilm activity.

Did the authors use negative and positive controls in the experiment ?

Author Response

 Minor comment:

Figure 4.

What was magnification used?

It is not visible in the photos, please complete the legend.

Re: We thank the reviewer for these comments that allowed us to improve the quality of Figure 4 legend,  the magnification for each image was added to the new version of the legend.

Figure 4. SEM image of HA 80000x (A), HA+C20 35000x (B), HA-Xccɸ1 68000x (C) and HA-Xccɸ1+C20 75000x (D). Scale bar is 1µm.

2.6. Long fatty acids anti-biofilm activity.

Did the authors use negative and positive controls in the experiment?

R: In the anti-biofilm experiment the untreated biofilm is used as the negative control (in this condition the maximum quantity of biofilm is obtained), as the positive control, the not inoculated well is use (in this condition no biofilm is formed).

Round 2

Reviewer 1 Report

1/ the results with the acids alone in figure 1B are very different from those in figure 1A, yet it seems that there was very little variability between trials in figure 1A, how do you explain this discrepancy?

so it seems that the variability of the effect of fatty acids is weaker after an 8-hour treatment.

2/ invalid qPCR 

you used a DNA treatment but you did not check for DNA contamination. This can be done by using RNA samples as template and using primers corresponding to the genomic region (non coding) flanking one gene

Moreover, you used "hcrC" but in Zaccardelli et al , it's "hrcC", it's not the same!

hrcC can be used for detection as in Zaccardelli et al but not as a reference gene in qRT-PCR. Please use atpD, rpoB, gyrA or gyrB : they are better candidates as they are housekeeping genes.

Author Response

Reply to reviewer 1

/ the results with the acids alone in figure 1B are very different from those in figure 1A, yet it seems that there was very little variability between trials in figure 1A, how do you explain this discrepancy?

so it seems that the variability of the effect of fatty acids is weaker after an 8-hour treatment.

RE:  In Figure 1A the treatment was performed for 8h and the acids resulted to be very effective against the Xcc biofilm and the difference between the different acids, in particular in case of c16, c18 and c20,  were not appreciable. Moreover in these conditions the test variability was low, we suppose that the experimental conditions caused an activity plateau, that is able to cover the difference between the different acids and the difference between the experimental replicates (low variability after an 8-hour treatment). For this reason, in the subsequent experiment, we reduced the time of treatment  (6h) to select a condition suitable to identify the most active acid to use. Indeed using a shorter time treatment, the effect of different fatty acids was different, and the  variability of the text was higher.

2/ invalid qPCR

you used a DNA treatment but you did not check for DNA contamination. This can be done by using RNA samples as template and using primers corresponding to the genomic region (non coding) flanking one gene

Re: we usually perform the control of DNA contamination using the RNA preparation as a template for PCR (using the primer selected for qPCR). We added a supplementary figure (Figure 3S) to show all the controls performed.

hrcC can be used for detection as in Zaccardelli et al but not as a reference gene in qRT-PCR. Please use atpD, rpoB, gyrA or gyrB : they are better candidates as they are housekeeping genes.

Re: we are in debt with the referee for this suggestion we repeated the qPCR experiments using rpoB as housekeeping.